# Bullying victimization among in-school adolescents in Sierra Leone: A cross-sectional analysis of the 2017 Sierra Leone Global School-Based Health Survey

**Augustus Osborne**[1]*, **Peter Bai James**[2,3], **Camilla Bangura**[1], **Samuel Maxwell Tom Williams**[1], **Jia Bainga Kangbai**[4], **Aiah Lebbie**[1]

1 Department of Biological Sciences, School of Environmental Sciences, Njala University, PMB, Freetown, Sierra Leone, 2 Faculty of Health, National Centre for Naturopathic Medicine, Southern Cross University, Lismore, Australia, 3 Faculty of Pharmaceutical Sciences, College of Medicine and Allied Health Sciences, University of Sierra Leone, Freetown, Sierra Leone, 4 Department of Environmental Health, School of Community Health Sciences, Njala University, PMB, Freetown, Sierra Leone

* augustusosborne2@gmail.com

**Data Availability Statement:** The dataset informing the findings of this study is publicly available. It can be freely accessed via the WHO

## Abstract

Adolescent bullying victimization is recognized as a public health and mental health problem in many countries. However, data on bullying victimization's prevalence and risk factors is scarce in sub-Saharan Africa Sierra Leone. This research aimed to determine bullying victimization prevalence and its associated factors among Sierra Leonean school-going adolescents. The Sierra Leone 2017 Global School-based Health Survey (GSHS) dataset was analyzed. The outcome variable was the respondent's self-report of bullying victimization ("How many days in the previous 30 days were you bullied?"). Descriptive, Pearson chi-square and binary logistic regression analyses were conducted. The regression analysis yielded adjusted odds ratios (aOR) with 95% confidence intervals (CIs) and a significance level of p 0.05. Bullying victimization was prevalent among 48.7% of the in-school adolescents. Adolescents who drank alcohol [aOR = 2.48, 95% CI = 1.50–4.10], who reported feelings of loneliness [aOR = 1.51, 95% CI = 1.07–2.14] and who had attempted suicide [aOR = 1.72, 95% CI = 1.03–2.87] were also more likely to be bullied. Also, school truancy [aOR = 1.53, 95% CI = 1.24–1.88] among teenagers was associated with an increased risk of being bullied. Our findings suggest that bullying is a widespread problem among Sierra Leonean school-aged youth, and alcohol drinking, loneliness, suicide attempt and school truancy are potential risk factors. In light of the aforementioned causes of bullying in schools, policymakers and school administrators in Sierra Leone need to develop and execute anti-bullying policies and initiatives that target the underlying risk factors of bullying among teenagers.

## Introduction

Approximately 32% of adolescents have experienced peer bullying at school on one or many occasions. The Middle East, North Africa, and sub-Saharan Africa exhibit the highest

NCD Microdata Repository https://extranet.who.int/ncdsmicrodata/index.php/catalog/GSHS.

**Funding:** The authors received no specific funding for this work.

**Competing interests:** The authors have declared that no competing interests exist.

prevalence of pupils reporting instances of bullying, whereas Central America, the Caribbean, and Europe demonstrate the lowest rates. [1–3] Bullying is a form of aggressive behaviour intended to cause harm or distress to the victim, and it can take many forms, including physical, verbal, and psychological. It is a pervasive problem that affects adolescents in many countries and can lead to various negative outcomes, such as depression, anxiety, low self-esteem, and even suicide [3].

Sierra Leone is a country in sub-Saharan Africa that has been identified as having a particularly high prevalence of bullying victimization among in-school adolescents [4]. This is concerning, as bullying victimization can seriously affect young people's physical and mental health. According to [5], the prevalence of bullying victimization among in-school adolescents in sub-Saharan Africa was 38.8%. Sierra Leone had the highest prevalence rate of bullying victimization at 54.6%, while Mauritius had the lowest rate at 22.2%. The study also found that socio-cultural, contextual, and socio-economic variations in the sub-region could contribute to bullying victimization. The post-conflict dynamics in Sierra Leone could trigger compulsive behaviours in school-going adolescents who may still have social exclusion through turbulent moments [6].

Understanding the risk factors contributing to bullying victimization among in-school adolescents in Sierra Leone is important for developing effective interventions to address the issue. A Ghanian study found that students engaging in physical fights, being attacked physically, are injured, lonely, attempted suicide, and were current users of marijuana associated with bullying victimization among in-school adolescents [5]. These findings suggest that interventions to address bullying victimization need to consider the broader social and cultural context in which it occurs and should involve families, communities, and schools [5].

In addition, to the negative effects on mental health, research has indicated that adolescents who skip school or drop out are more likely to become victims of bullying and this can have long-term consequences for their future opportunities and well-being.[6], Addressing bullying victimization in schools is therefore important for young people's mental health, academic success, and future prospects. Many interventions can be used to address bullying victimization among in-school adolescents. These include school-based programs that promote positive behaviours, reduce aggressive behaviour, and support victims and their families. Parental involvement in these programs can also be effective in reducing bullying victimization. Peer support programs, such as peer mentoring and peer counselling, can also effectively reduce bullying victimization and improve mental health outcomes for victims [6].

Child and adolescent mental health in Sierra Leone have received limited attention, but there have been endeavours to enhance child and adolescent mental healthcare in a low resource country. The school's curriculum incorporates a comprehensive approach to health education. The curriculum encompasses various subjects, including sexual and reproductive health, hygienic practises, nutritional education, substance misuse prevention, and mental health awareness. Acknowledging the significance of mental health, school health services integrate counselling and support provisions aimed at assisting adolescents in managing stress, anxiety, and other mental health difficulties. Schools often coordinate and implement routine health campaigns and awareness programmes. These campaigns aim to tackle concerns such as HIV/AIDS prevention, malaria control, and vaccine initiatives, with the goal of equipping adolescents with comprehensive knowledge regarding preventive measures [7].

In Sierra Leone, no studies have examined the prevalence of bullying or its effects on its victims. More research is needed to understand better the correlates of bullying victimization among Sierra Leonean in-school adolescents; this will assist in guiding public health initiatives to reduce the incidence of bullying victimization in school settings. This research sought to determine bullying victimization prevalence and its associated factors among Sierra Leonean school-going adolescents.

## Materials and methods

### Sample and procedure

The 2017 Global School Health Survey (GSHS) in Sierra Leone provided a cross-sectional dataset that we were able to utilize [8]. The Sierra Leone data collection uses a two-stage cluster sampling methodology to collect an accurate cross-section of the country's 10–19-year-old student population. The first step includes picking schools with a probability proportional to students' enrolment. The second stage is randomly picking classes such that every student has an equal chance of being selected. The response rates for schools in Sierra Leone's GSHS were 94%, and the student response rate was 87% [8]. Our study aligns with STROBE guidelines for observational studies (S1 Checklist). The GSHS dataset provides a nationally representative information on health behaviours and protective factors that are predisposing factor morbidity and mortality among adolescents in middle- and lower-income countries [9]. The goal is to support school health and youth health programs and policies worldwide.

GSHS collects data on the following health behaviours—alcohol use, tobacco use, dietary behaviours, drug use, hygiene, mental health and physical activity and violence and unintentional injury. Also, collects data on sexual behaviours such as HIV infection, other sexually transmitted infections, and unintended pregnancy. Protective factors include peer and parental support [9].

### Outcome variable

The study's dependent variable was bullying victimization. The original question was, "How many days in the last 30 days have you been bullied?" Answers 1 represented no days, 2 represented 1 or 2 days, 3 represented 3 to 5 days, 4 represented 6 to 9 days, 5 represented 10 to 19 days, 6 represented 20 to 29 days, and 7 represented all 30 days. In this analysis, yes/no answers were the only possible outcomes. We classified adolescents as "Yes" if they reported being bullied at least once in the last 30 days and as "No" if they reported never being bullied (1 = 0 days). The replies were classified based on research by [5,10].

### Explanatory variables

The explanatory factors employed in the analysis were selected because of their availability in the GSHS dataset and historical connections with the dependent variables (e.g., [5,10]). Age, gender, grade, truancy, alcohol usage, suicidal thoughts, suicide attempt, loneliness, anxiety, and being bullied were some demographic and health risk characteristics examined. The protective factors are (peer support, close friends, parental or guardian supervision, parental or guardian bonding, and parental or guardian connectedness). Our research variables have been widely utilized and verified in prior research [5,10]. (Table 1) displays a thorough explanation of the variables and the recoded replies.

### Statistical analyses

SPSS version 28 was used to analyze the data. Descriptive, Pearson chi-square and binary logistic regression analyses were carried out. The frequency distributions and percentages utilized to illustrate the categories' distributions were descriptive. Second, the correlation between bullying victimization (the outcome variable) and the explanatory factors was analyzed using the Pearson chi-square test. We used the complex sampling command on SPSS to account for weighting and complex sampling design. All variables were included in the logistic regression, but only those with a significant connection (p 0.05) were used. The regression analysis findings were shown as odds ratios (aOR) with 95% confidence intervals (CIs). The p-value needed to be less

**Table 1. Study variables.**

| Variable | Survey question | Original response options | Recoded |
|---|---|---|---|
| **Age** | How old are you? | 11–18 years (coded categorically) | 1–4 = ≤14years; 5–8 = ≥15years |
| **Sex** | What is your sex? | 1 = male; 2 = female | 1 = male; 0 = female |
| **Grade** | In what grade are you? | 1 Jnr Sec (JSS) 2 to 5 Snr Sec (SSS) 3 | N/A |
| **Current alcohol use** | During the past 30 days, how many days did you have at least one drink containing alcohol? | 1 = 0 days to 7 = All 30 days | 1 = 0, 2–7 = 1 |
| **Cannabis use** | During your life, how many times have you used marijuana | 1 = 0 times to 5 = 20 or more times | 1 = 0 and 2–5 = 1 |
| Suicidal Ideation | During the past 12 months, did you ever seriously consider attempting suicide? | Yes = 1; no = 2 | Yes = 1 and no = 0 |
| **Suicidal plan** | During the past 12 months, did you make a plan About how you would attempt suicide? | Yes = 1; no = 2 | Yes = 1 and no = 0 |
| **Suicidal attempt** | During the past 12 months, how many times did you actually attempt suicide? | 1 = 0 times to 5 = 6 times or more | 1 = 0 and 2–5 = 1 |
| **Close friends** | How many close friends do you have? | 1 = 0 and 4 = 3 or more | 1 = 0 and 2–4 = 1 |
| **Loneliness** | During the past 12 months, how often have you felt lonely? | 1 = never to 5 = always | 1–3 = 0 and 4–5 = 1 |
| **Anxiety** | During the past 12 months, how often have you been so worried about something that you could not sleep at night? | 1 = never to 5 = always | 1–3 = 0 and 4–5 = 1 |
| **Bullied** | During the past 30 days, how many days were you bullied? | 1 = 0 days to 7 = all 30 days | 1 = 0 and 2–7 = 1 |
| School truancy | During the past 30 days, how many days did you miss classes or school without permission? | 1 = 0 days to 5 = 10 or more days | 1 = 0 and 2–5 = 1 |
| **Peer support** | During the past 30 days, how often were most of the students in your school kind and helpful? | 1 = never to 5 = always | 1–3 = 0 and 4–5 = 1 |
| **Parental monitoring** | During the past 30 days, how often did your parents or guardians check to see if your homework was done? | 1 = never to 5 = always | 1–3 = 0 and 4–5 = 1 |
| **Parental understanding** | During the past 30 days, how often did your parents or guardians understand your problems and worries? | 1 = never to 5 = always | 1–3 = 0 and 4–5 = 1 |
| **Parental bonding** | During the past 30 days, how often did your parents or guardians really know what were you doing with your free time? | 1 = never to 5 = always | 1–3 = 0 and 4–5 = 1 |
| Parental intrusion of privacy | During the past 30 days, how often did your parents or guardians go through your things without your approval? | 1 = never to 5 = always | 1–3 = 0 and 4–5 = 1 |

than 5% to be statistically significant. We assess multicollinearity among independent variables using the variance inflation factor. Listwise deletion was used to address missing data.

## Ethics statement

We got permission to utilise the data from the global school-based health survey programme. Since we analyzed a secondary dataset in which the participants' personal information had been removed, we didn't need to get formal ethical permission to perform this research. However, prior to conducting the surveys, ethical permission was acquired from the Ministry of Health and Sanitation in Sierra Leone.

## Results

A total of 2,798 in-school adolescents in Sierra Leone was analyzed in the present study. As seen in (Fig 1), a significant percentage of Sierra Leonean adolescents have been bullied while

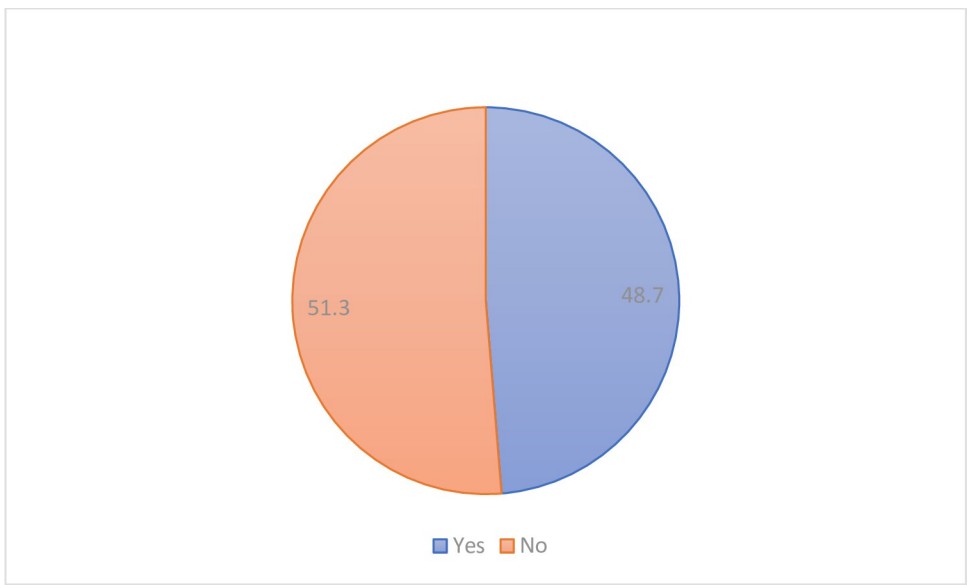

**Fig 1. Prevalence of bullying victimisation among adolescents in Sierra Leone.**

attending school. The percentage of Sierra Leonean adolescents who had been bullied while attending school was 48.7%.

Bullying victimization among Sierra Leonean adolescents in school is shown in (Table 2), along with a bivariate analysis. The percentage of Sierra Leonean adolescents bullied while attending school was 48.7%. Victimization by bullies varied greatly by factors such as loneliness, attempts at suicide, substance abuse, and school absences. Significant differences across variables all had p-values of less than 0.05. Males accounted for the largest proportion of bullying victims (53.1%), lonely school-aged children (23%), and those who had attempted suicide (24.5%). Adolescents who regularly partake in the usage of both alcohol (19.2%) and cannabis (6.3%) are disproportionately likely to be bullied while at school. Also, students who skipped class were likelier to become bully victims than those who stayed in class (41.1%). The variables that did not show statistically significant associations with bullying victimization include age, sex, grade, anxiety, suicidal ideation, suicidal plan, close friends, peer support, and parental support, with p-values greater than 0.05.

The results of the multivariate logistic regression are shown in (Table 3). Adolescents who drank alcohol [aOR = **2.48,** 95% CI = **1.50–4.10**] were more likely to be bullied. Furthermore, teenagers who reported feelings of loneliness [aOR = **1.51,** 95% CI = **1.07–2.14**] were more likely to report being bullied than their less lonely peers. Adolescents who had already attempted suicide [aOR = **1.72,** 95% CI = **1.03–2.87**] were also more vulnerable to being bullied. Similarly, school truancy [aOR = **1.53,** 95% CI = **1.24–1.88**] among teenagers was associated with an increased risk of being bullied. The variables that did not show statistically significant associations with bullying victimization include age, sex, grade, cannabis use, anxiety, suicidal ideation, suicidal plan, close friends, peer support, and parental support.

## Discussion

School-aged adolescents in Sierra Leone were found to have a prevalence of 48.7% of having been bullied on one or more days over the preceding 30 days, with a prevalence of 53.1% among boys and a prevalence of 46.9% among girls. Research on bullying in eight African

**Table 2. Bivariate analysis of bullying victimization across the background characteristics in Sierra Leone.**

| Characteristics | Variables | Bullied | | X², p-value |
|---|---|---|---|---|
| | | Yes n (%) | No n (%) | |
| Age | ≤14yrs | 489(36.4) | 484(34.9) | 0.63, p = 0.583 |
| | ≥15years | 859(63.6) | 948(65.1) | |
| Sex | Male | 633(53.1) | 625(50.3) | 2.20, p = 0 .129 |
| | Female | 691(46.9) | 793(49.7) | |
| Grade | JSS | 902(72.9) | 873(68.6) | 6.19, p = 0 .338 |
| | SSS | 446(27.1) | 554(31.4) | |
| Alcohol use | Yes | 265(19.2) | 128(8.2) | 68.21, p<0.001 |
| | No | 978(80.8) | 1246(98.1) | |
| Cannabis use | Yes | 103(6.3) | 35(2.5) | 24.06, p = 0.002 |
| | No | 1185(93.7) | 1361(97.5) | |
| Loneliness | Less Lonely | 1008(77.0) | 1194(83.9) | 21.25,p = 0 .020 |
| | Lonelier | 328(23.0) | 238(16.1) | |
| Anxiety | Less or no anxiety | 1058(79.9) | 1206(83.0) | 4.54, p = 0.256 |
| | Anxiety | 292(20.1) | 228(17.0) | |
| Suicidal Ideation | Yes | 222(16.6) | 172(11.9) | 12.62, P = 0.213 |
| | No | 1075(83.4) | 1225(88.1) | |
| Suicidal plan | Yes | 229(16.2) | 224(15.7) | 0 .16, P = 0.862 |
| | No | 1084(83.8) | 1198(84.3) | |
| Suicidal attempt | Yes | 335(24.5) | 206(14.1) | 47.76, p<0.001 |
| | No | 999(75.5) | 1220(85.9) | |
| Close friends | No close friends | 130(9.9) | 119(8.6) | 1.26, p = 0.452 |
| | Close friends | 1192(90.1) | 1290(91.4) | |
| School truancy | Yes | 555(41.1) | 437(29.2) | 42.93, p<0.001 |
| | No | 786(58.9) | 991(70.8) | |
| Peer support | Yes | 393(28.2) | 461(31.0) | 2.69, p = 0.160 |
| | No | 944(71.8) | 961(69.0) | |
| Parental Support | Yes | 936(75.1) | 1077(78.4) | 4.05, p = 0.216 |
| | No | 347(24.9) | 308(21.6) | |

countries [11] found rates that were higher than those found in this study, ranging from 25% in Tanzania to 63% in Zambia. In comparison, research on bullying in nine developing countries [12] found lower rates, with 42% of males and 39% of females reporting being bullied (at least once in the preceding 30 days). Differences in survey years, sample sizes, and contextual factors such as differences in schools, neighbourhoods, and cultures may all explain these discrepancies [13].

In line with the findings of earlier research [11,14–16], this one indicated that boys were more likely to have been bullied than girls.

Bullying was more common among those who used drugs like cannabis and alcohol. According to a growing body of national and international research, adolescents who are bullied are at increased risk for drug usage and misuse [17]. Previous research [18,19] has shown that teenage drug usage increases the likelihood of being bullied as a victim. Although several researchers have shown that substance addiction increases a teen's risk of being bullied [18,20], most of these investigations have focused on cigarettes. As a result, this study's discovery that cannabis and alcohol usage raise the risk of being bullied lays the stage for future research aiming to replicate this study.

**Table 3. Multivariable regression analysis of predictors of bullying victimization among in-school adolescents in Sierra Leone.**

| Characteristics | Variables | aOR (95%CI) |
| --- | --- | --- |
| Age | ≤14yrs | 1 |
| | >15years | 0.89(0.65–1.22) |
| Sex | Male | 1.09(0.89–1.34) |
| | Female | 1 |
| Grade | JSS | 1 |
| | SSS | 0.73(0.42–1.26) |
| Alcohol use | Yes | **2.48(1.50–4.10)** * |
| | No | 1 |
| Cannabis use | Yes | 1.16(0.53–2.54) |
| | No | 1 |
| Loneliness | Less Lonely | 1 |
| | Lonelier | **1.51(1.07–2.14)** |
| Anxiety | Less or no anxiety | 1 |
| | Anxiety | 1.00(0.69–1.46) |
| Suicidal Ideation | Yes | 1.11(0.67–1.83) |
| | No | 1 |
| Suicidal plan | Yes | 0.77(0.45–1.32) |
| | No | 1 |
| Suicidal attempt | Yes | **1.72(1.03–2.87)** |
| | No | 1 |
| Close friends | No close friends | 1 |
| | Close friends | 0.95(0.60–1.52) |
| School truancy | Yes | **1.53(1.24–1.88)** |
| | No | 1 |
| Peer support | Yes | 1 |
| | No | 1.07(0.84–1.36) |
| Parental Support | Yes | 1 |
| | No | 1.07(0.78–1.50) |

Researchers have shown a strong correlation between bullying victimization and subsequent suicide attempts [21,22]. We found that students who had tried suicide were more likely to be bullied at school, confirming these previous results. This result exemplifies the fact that not only can suicidal attempts predict bullying victimization, but victimization by bullies may also increase the risk of suicide attempts. Suicide attempts are more common among bully victims because of the sentiments of isolation and rejection that they experience, as stated in Opperman et al [23]. Suicide thoughts and attempted suicide are further exacerbated by poor self-esteem [24] and impaired self-worth [25]. Our results show that contrary to what has been found in the existing research, the risk of a teenager experiencing bullying at school may be predicted by whether or not they had attempted suicide.

The likelihood of being bullied is higher for those who often feel alone. This conclusion is congruent with the findings from Dembo and Gulledge [26], which showed that teenage victims of bullying were more prone to experience despair and loneliness. Because adolescence is a time of change and growth, it is natural for teenagers to seek community and friendship. When this isn't the case, teenagers instead experience isolation, which make them more vulnerable to many forms of bullying [27].

In addition, this research found no association between being closely supervised by parents and having close friends [16,28]. The research found that school truancy is associated with a higher risk of being bullied. This can be addressed with Check and Connect as a school-based intervention program that encourages adolescents to attend class on a consistent basis and addresses issues including absenteeism and a lack of parental or guardian connection [29,30]. In the checking phase, we look for warning signals, such as missing school, that might indicate a problem. The connecting part of the concept uses a personalized mentor/monitor setup to establish a lasting bond with the student, their loved ones, and the school community [30].

## Policy and practice implication

In order to reflect the experiences of each demographic subgroup more accurately, anti-bullying initiatives in schools should focus on the unique challenges that students face at each grade level. Other school-based involvement should include the fact that many teenagers, particularly boys, experience physical forms of bullying victimization e.g., physical fights, are assaulted, display signs of loneliness, and may try suicide. School administrators and policymakers should work together to improve school environment by reviving and developing new anti-bullying programs and improving the effectiveness of current ones.

## Study limitations

The research conducted in Sierra Leone in 2017 employed a cross-sectional study design, which limits our capacity to establish causal relationships between the dependent and explanatory variables. Additionally, it is crucial to acknowledge that the scope of our findings is restricted to teenagers who are enrolled in schools within Sierra Leone. Future study should place a higher emphasis on investigating the experiences and behaviour exhibited by adolescents in both educational and non-educational environments. The presence of recall bias in the data is a potential concern as it is influenced by the dependence on self-reported responses.

## Conclusions

Our study reported an alarmingly high levels of suicidal attempt among adolescents in Sierra Leone. This study also lends credence to the idea that bullying is a widespread problem among Sierra Leonean school-aged youth. Sierra Leonean school-going adolescents who are alcohol users, were lonely, had history of suicide attempt and truancy were more likely to be bullied. In light of the aforementioned risk factors of bullying in schools, policymakers and school administrators in Sierra Leone need to develop and execute anti-bullying policies and initiatives that target the underlying causes of bullying among teenagers. They should also increase awareness on suicide in Sierra Leone. This can be done through public campaigns, school campaign and community outreach.

## Supporting information

**S1 Checklist. STROBE statement—checklist of items that should be included in reports of observational studies.**
(DOCX)

## Acknowledgments

The authors would like to express their appreciation to the World Health Organization for providing them with the data used in this study. Participants in the 2017 GSHS in Sierra Leone are also thanked, as are the country's Education and Health Ministries.

## Author Contributions

**Conceptualization:** Augustus Osborne, Peter Bai James, Camilla Bangura, Samuel Maxwell Tom Williams, Jia Bainga Kangbai, Aiah Lebbie.

**Data curation:** Augustus Osborne.

**Formal analysis:** Augustus Osborne, Peter Bai James.

**Methodology:** Augustus Osborne.

**Supervision:** Augustus Osborne, Peter Bai James, Camilla Bangura, Samuel Maxwell Tom Williams, Jia Bainga Kangbai, Aiah Lebbie.

**Validation:** Augustus Osborne, Peter Bai James, Camilla Bangura, Samuel Maxwell Tom Williams, Jia Bainga Kangbai, Aiah Lebbie.

**Visualization:** Augustus Osborne, Peter Bai James, Camilla Bangura, Samuel Maxwell Tom Williams, Jia Bainga Kangbai, Aiah Lebbie.

**Writing – original draft:** Augustus Osborne.

**Writing – review & editing:** Peter Bai James, Camilla Bangura, Samuel Maxwell Tom Williams, Jia Bainga Kangbai, Aiah Lebbie.

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
