## [Decision Letter · Decision Letter 0]

12 Sep 2023

PGPH-D-23-01085

Bullying Victimization among in-School Adolescents in Sierra Leone: Analysis of the 2017 Sierra Leone Global School-Based Health Survey.

Dear Augustus Osborne

Thank you for submitting your manuscript to PLOS Global Public Health. After careful consideration, we feel that it has merit but does not fully meet PLOS Global Public Health’s publication criteria as it currently stands. Therefore, we invite you to submit a revised version of the manuscript that addresses the points raised during the review process.

A rebuttal letter that responds to each point raised by the reviewer(s). You should upload this letter as a separate file labeled 'Response to Reviewers'.A marked-up copy of your manuscript that highlights changes made to the original version. You should upload this as a separate file labeled 'Revised Manuscript with Track Changes'.An unmarked version of your revised paper without tracked changes. You should upload this as a separate file labeled 'Manuscript'.

We look forward to receiving your revised manuscript.

Kind regards,

Suvarna Jyothi Kantipudi, MBBS,DPM,MD

Academic Editor

Journal Requirements:

1. Please provide separate figure files in .tif or .eps format only and remove any figures embedded in your manuscript file. Please also ensure all files are under our size limit of 10MB.

Reviewer 1:

The authors may consider reviewing the language of the article. At certain places the sentence structuring can be better to derive proper meaning.

2. The Ethics review for secondary data analysis has been skipped. a waiver should have been obtained from Institutional ethics committee.

3. In the first half of the paper there is inadequate clarity on temporality - thus it seems to suggest that the potential effects of bullying are rather the risk factors / predisposing factors for bullying - e.g. Alcohol use. This needs correction in language and sentence framing.

4. At many places only those factors which were statistically significant at 5% alpha error have been reported creating a kind of reporting bias. e.g. lack of association with suicidal ideation but presence of statistically significant association with suicidal attempt needs reporting and explanation, including limitation.

5. Very high levels of suicidal attempt have been reported. However, the same has not received much attention in conclusion and recommendation to authorities.

6. Figure needs proper labelling and is creating confusion with putting frequency and percentage in a single figure. consider revising.

The article needs thorough revising based on the recommendations above.

Reviewer 2:

Introduction:

The introductory lines " Bullying victimization among in-school adolescents is a serious public health issue that has

gained increasing attention in recent years." It is better to cite sources which provide data on bullying among adolescents rather than research articles.

Description of school health services addressing adolescent health in Sierra Leone region has to be mentioned in the introductory section in order to provide overview.

Methodology:

In the explanatory variable, a detailed description of GSHS dataset has to be provided.

Results:

Results section to provide data on total number of adolescents included in the analysis.

Discussion:

In the limitation section, the authors commented on various limitations, however these were not potrayed in methodology section. Hence, sampling strategy has to be revised and provide detailed description rather than few lines

Reviewers' comments:

Reviewer's Responses to Questions

**Comments to the Author**

1. Does this manuscript meet PLOS Global Public Health’s publication criteria? Is the manuscript technically sound, and do the data support the conclusions? The manuscript must describe methodologically and ethically rigorous research with conclusions that are appropriately drawn based on the data presented.

Reviewer #1: Yes

Reviewer #2: No

Reviewer #3: Yes

2. Has the statistical analysis been performed appropriately and rigorously?

Reviewer #1: Yes

Reviewer #2: No

Reviewer #3: Yes

3. Have the authors made all data underlying the findings in their manuscript fully available (please refer to the Data Availability Statement at the start of the manuscript PDF file)?

Reviewer #1: Yes

Reviewer #2: No

Reviewer #3: Yes

4. Is the manuscript presented in an intelligible fashion and written in standard English?

Reviewer #1: Yes

Reviewer #2: No

Reviewer #3: Yes

5. Review Comments to the Author

Reviewer #1: I would like to thank the editor for providing me the opportunity to review the current article.

Specific comments to authors:

Introduction:

The introductory lines " Bullying victimization among in-school adolescents is a serious public health issue that has

gained increasing attention in recent years." It is better to cite sources which provide data on bullying among adolescents rather than research articles.

Description of school health services addressing adolescent health in Sierra Leone region has to be mentioned in the introductory section in order to provide overview.

Methodology:

In the explanatory variable, a detailed description of GSHS dataset has to be provided.

Results:

Results section to provide data on total number of adolescents included in the analysis.

Discussion:

In the limitation section, the authors commented on various limitations, however these were not potrayed in methodology section. Hence, sampling strategy has to be revised and provide detailed description rather than few lines

Reviewer #2: Introduction:

This portion fails to justify the rationale of the current study i.e. why this study was carried out and what significance it could make to the existing body of knowledge. More baseline work is required e.g. how Pakistani pregnant women would differ from women belonging to other parts of the world in developing pregnancy related anxiety. Cultural factors of infertility and the expectation of a male child may also be discussed.

Method:

More detail on the CBT practitioners and their expertise is required.

Results:

Either hypotheses or Research Questions or Objectives should be formulated prior to calculating the results. Proper statistical procedures must be performed afterwards.

Table should be in align with standard formats e.g. APA.

Reviewer #3: The study looking at prevalence and associated factors of bullying victimisation in Sierra Leone looks at the database available from 2017 and performs analysis to find out significant factors.

The study looks at an important aspect that affects adolescent health especially mental health.

The study as a secondary analysis of a quantitative survey which is nearly six years old has its imitations when deriving conclusions from the same.

The article has been well structured and written and certain edits will make it adequately rigorous for publishing in PLOS GPH.

1. The authors may consider reviewing the language of the article. At certain places the sentence structuring can be better to derive proper meaning.

2. The Ethics review for secondary data analysis has been skipped. a waiver should have been obtained from Institutional ethics committee.

3. In the first half of the paper there is inadequate clarity on temporality - thus it seems to suggest that the potential effects of bullying are rather the risk factors / predisposing factors for bullying - e.g. Alcohol use. This needs correction in language and sentence framing.

4. At many places only those factors which were statistically significant at 5% alpha error have been reported creating a kind of reporting bias. e.g. lack of association with suicidal ideation but presence of statistically significant association with suicidal attempt needs reporting and explanation, including limitation.

5. Very high levels of suicidal attempt have been reported. However, the same has not received much attention in conclusion and recommendation to authorities.

6. Figure needs proper labelling and is creating confusion with putting frequency and percentage in a single figure. consider revising.

The article needs thorough revising based on the recommendations above.

6. PLOS authors have the option to publish the peer review history of their article (what does this mean?). If published, this will include your full peer review and any attached files.

**Do you want your identity to be public for this peer review?** For information about this choice, including consent withdrawal, please see our Privacy Policy.

Reviewer #1: No

Reviewer #2: No

Reviewer #3: No

---

## [Decision Letter · Decision Letter 1]

29 Nov 2023

Bullying Victimization among in-School Adolescents in Sierra Leone: Analysis of the 2017 Sierra Leone Global School-Based Health Survey.

PGPH-D-23-01085R1

Dear Augustus Osborne,

We are pleased to inform you that your manuscript 'Bullying Victimization among in-School Adolescents in Sierra Leone: Analysis of the 2017 Sierra Leone Global School-Based Health Survey.' has been provisionally accepted for publication in PLOS Global Public Health.

Best regards,

Lola Kola, PhD

Academic Editor

The total sample should be included in the abstract and mentioned in the methods too.

Reviewer Comments (if any, and for reference):

Reviewer's Responses to Questions

**Comments to the Author**

1. If the authors have adequately addressed your comments raised in a previous round of review and you feel that this manuscript is now acceptable for publication, you may indicate that here to bypass the “Comments to the Author” section, enter your conflict of interest statement in the “Confidential to Editor” section, and submit your "Accept" recommendation.

Reviewer #1: All comments have been addressed

2. Does this manuscript meet PLOS Global Public Health’s publication criteria? Is the manuscript technically sound, and do the data support the conclusions? The manuscript must describe methodologically and ethically rigorous research with conclusions that are appropriately drawn based on the data presented.

Reviewer #1: Yes

3. Has the statistical analysis been performed appropriately and rigorously?

Reviewer #1: Yes

4. Have the authors made all data underlying the findings in their manuscript fully available (please refer to the Data Availability Statement at the start of the manuscript PDF file)?

Reviewer #1: Yes

5. Is the manuscript presented in an intelligible fashion and written in standard English?

Reviewer #1: Yes

6. Review Comments to the Author

Reviewer #1: (No Response)

7. PLOS authors have the option to publish the peer review history of their article (what does this mean?). If published, this will include your full peer review and any attached files.

**Do you want your identity to be public for this peer review?** For information about this choice, including consent withdrawal, please see our Privacy Policy.

Reviewer #1: No
